# Computerized Texture Analysis of Optical Coherence Tomography Angiography of Choriocapillaris in Normal Eyes of Young and Healthy Subjects

**DOI:** 10.3390/cells11121934

**Published:** 2022-06-15

**Authors:** Asadolah Movahedan, Phillip Vargas, John Moir, Gabriel Kaufmann, Lindsay Chun, Claire Smith, Nathalie Massamba, Patrick La Riviere, Dimitra Skondra

**Affiliations:** 1J. Terry Ernest Ocular Imaging Center, Department of Ophthalmology and Visual Science, The University of Chicago, Chicago, IL 60637, USA; asadolah.movahedan@uth.tmc.edu (A.M.); john.moir@uchospitals.edu (J.M.); gabriel.kaufmann@uchospitals.edu (G.K.); lindsay.chun@uchospitals.edu (L.C.); clairesylvestersmith@gmail.com (C.S.); nmassamba@bsd.uchicago.edu (N.M.); 2Department of Ophthalmology and Visual Science, University of Texas Health Sciences, Houston, TX 77030, USA; 3Department of Radiology, The University of Chicago, Chicago, IL 60637, USA; vargasp@uchicago.edu (P.V.); pjlarivi@uchicago.edu (P.L.R.)

**Keywords:** computerized texture analysis, optical coherence tomography angiography, choriocapillaris

## Abstract

Computerized texture analysis uses higher-order mathematics to identify patterns beyond what the naked eye can recognize. We tested its feasibility in optical coherence tomography angiography imaging of choriocapillaris. Our objective was to determine sets of parameters that provide coherent and consistent output when applied to a homogeneous, healthy group of patients. This observational cross-sectional study involved 19 eyes of 10 young and healthy Caucasian subjects. En-face macular optical coherence tomography angiography of superficial choriocapillaris was obtained by the RTVue-XR Avanti system. Various algorithms were used to extract texture features. The mean and standard deviation were used to assess the distribution and dispersion of data points in each metric among eyes, which included: average gray level, gray level yielding 70% threshold and 30% threshold, balance, skewness, energy, entropy, contrast, edge mean gradient, root-mean-square variation, and first moment of power spectrum, which was compared between images, showing a highly concordant homology between all eyes of participants. We conclude that computerized texture analysis for en-face optical coherence tomography angiography images of choriocapillaris is feasible and provides values that are coherent and tightly distributed around the mean in a homogenous, healthy group of patients. Homology of blob size among subjects may represent a “repeat pattern” in signal density and thus a perfusion in the superficial choriocapillaris of healthy young individuals of the same ethnic background.

## 1. Introduction

The choriocapillaris (CC), the superficial layer of the choroidal vasculature, is of great clinical interest and has been demonstrated to show microscopic pathological changes in patients with prevalent retinal diseases [1,2,3,4,5,6,7]. However, this important layer has been out of the reach of previous in vivo retinal imaging modalities due to its complex vascular pattern, deep anatomical location, and the light scattering properties of the retinal pigment epithelium [8,9,10,11]. Moreover, angiographic techniques using fluorescein and indocyanine dyes have been restricted in their success due to the fenestration of the CC vasculature and resulting excessive leakage of dye.

Optical coherence tomography angiography (OCTA) is a dye-less, in vivo imaging technique that detects blood flow by comparing signal intensity in sequential scans and thus offers a novel technology for imaging the CC layer. En-face OCTA imaging technology has led to significant discoveries in CC flow and density in normal subjects [12,13] and has demonstrated changes in capillary density and CC flow associated with pathologies such as age-related macular degeneration and diabetic retinopathy [14,15,16,17,18,19,20]. Data to this end are encouraging but limited. Despite advances, analysis of OCTA imaging so far has been mostly limited to binary discrimination of lighter versus darker pixels, corresponding to faster versus slower or absent blood flow, respectively [15,16,17,18,19,20]. In the past few years, there has been a great interest to explore ultrahigh speed OCTA technology, and many studies have found new clinical applications for this modality [15,16,21,22,23,24,25,26].

Computerized texture analysis has been previously validated for use in medical imaging with analyses of breast mammography, radiography, computerized tomography, and vascular ultrasound [27,28,29,30,31,32,33,34,35,36,37,38,39]. Computerized texture analysis has not been previously explored for analysis of OCTA images, and evidence suggests that it can offer discrimination of spectral data beyond the naked eye through comparison of light and dark to include second-order statistical information regarding pixel correlation, skew, and coarseness [40,41]. Furthermore, quantitative analysis of OCT/OCTA imaging has been used in wide-ranging fields in biology and medicine for the characterization of imaging features found to correlate with histology, suggesting a role for texture analysis of OCTA images of the choriocapillaris [42,43,44,45]. Given the clinical significance of the CC and the current challenges in analyzing and interpreting OCTA images of this layer with widely available technology, we investigated the feasibility of applying computerized texture analysis to en-face OCTA imaging of the CC of normal subjects to extract quantitative metrics representing the texture of vascular patterns/perfusion.

## 2. Materials and Methods

This is an observational cross-sectional study in which 19 eyes of 10 young healthy Caucasian subjects with a minimal refractive error (spherical equivalent (SE) of −1.0 to +1.0 diopters) were enrolled. Participants had no known retinal pathology or systemic illnesses. The research study and design were approved by The University of Chicago’s Institutional Review Board. Informed consent was obtained from all participants. All experiments were performed in accordance with the declaration of Helsinki.

The RTVue-XR Avanti OCTA system (Optovue Inc., Fremont, CA, USA) with split-spectrum amplitude-decorrelation angiography (SSADA) software (Fremont, CA, USA) was used to obtain en-face 3 × 3 mm OCTA images of the superficial CC using automated segmentation. SSADA enhances the signal to noise ratio of flow, enabling the visualization of retinal and choroidal vasculature. The software also included 3D projection artifact removal (PAR). The subfoveal choroidal thickness of each eye was measured using the built-in caliper tool from the RPE to the sclerochoroidal boundary. The blood flow area (BFA) of a 3 mm diameter circle overlaid on the CC was measured using built-in AngioAnalytics software (Fremont, CA, USA) and represented the percentage of pixels corresponding to blood flow.

Images were analyzed using computerized texture analysis algorithms to extract texture features in five categories:Absolute value of gray levels: including average gray level, gray level yielding 70% threshold, and gray level yielding 30% threshold of the area under the histogram.Gray-level histogram analysis, including balance and skewness parameters.Spatial relationship among gray levels: including coarseness, contrast, energy, and entropy parameters.Edge frequency: defined as mean gradient as a function of distance between pixels.Fourier transform analysis, including root-mean-square variation (RMS) and first moment of power spectrum (FMP).

The mean and standard deviation from the mean were used to assess the distribution and dispersion of data points in each metric among eyes. Established choroidal biomarkers, including choroidal thickness and 3 mm CC flow, were correlated with textural parameters using Pearson’s correlation coefficient. A summary of the algorithms used for measurement of textural parameters is included in Appendix A.

## 3. Results

### 3.1. Demographics

To test feasibility, we defined a homogenous group of subjects consisting of 19 eyes of 10 young, healthy Caucasians (3 men and 7 women) with an average age of 24.7 ± 2.5 years with no or negligible refractive errors (range SE: −1.0 D to +1.0 D).

### 3.2. Textural Features

The mean and standard deviation from the mean were measured to assess the distribution and dispersion of data points for each metric among eyes (Table 1). Pearson’s correlation coefficient was used to assess the relationship between the subfoveal choroidal thickness and 3 mm CC flow and the various textural parameters (Table 1). Both of these established biomarkers were moderately positively or negatively correlated with the majority of the textural features. Using algorithms to extract desired textural parameters could be visually demonstrated by using different filters (Figure 1). Filters can extract patterns such as blobs and local binary patterns on en-face 3 × 3 mm macular OCTA images of the CC. This indicates that a two-dimensional image of the CC can be broken down to elements, allowing an observer to discern hidden patterns in otherwise similar images.

Blob size distribution denoting a pattern of homology in healthy young adults of a single race is shown in Figure 2A. Blob size distribution with highly concordant homology between all eyes may represent a repeated pattern of signal density in the superficial CC of healthy young individuals of the same racial background. Figure 2B depicts a normalized Fourier transform plot of all eyes, signifying that despite minor differences, there is a high structural similarity in the textural patterns of the CC in healthy young adults. Figure 3 and Figure 4 demonstrate a tight distribution of data points in each metric around the mean: gray level histogram analysis consisting of measures of balance and skewness (Figure 3A,B); absolute value of gray levels in OCTA images of the CC in each eye measured by average gray level, gray level with 30% or 70% threshold (Figure 3C–E), and spatial relationship among gray levels measured by energy, entropy, and contrast (Figure 3F–H); Fourier transform analysis of parameters included Root mean square (RMS) variation and First Moment of Power (FMP) spectrum (Figure 4A,B); and finally Edge frequency analysis (Figure 4C).

## 4. Discussion

It is known that the human visual system has difficulty discriminating higher-order texture information, particularly the spectral frequency properties of an image such as its coarseness or pattern. The en-face OCTA image of the CC is a complex image containing an enormous amount of information regarding the pattern of blood flow in the CC; current analysis methods, which mainly quantify flow voids or deficits based on a black–white pixel density, are unable to interpret this information fully and with great accuracy.

In this study, sets of mathematical calculations referred to as computerized texture analysis were applied to en-face OCTA images of the CC in healthy subjects. The main objective was to determine which one of the previously used sets of parameters provided coherent and consistent output when applied to OCTA images of a homogeneous group of subjects.

Computerized texture analysis utilizing similar algorithms and parameters as in our study has emerged as a valuable tool in imaging analysis in other organs. The included parameters, notably entropy [37,38], energy [38], and contrast [3,38], have proven successful for the discrimination of healthy versus at-risk or diseased tissue in various specialties and imaging modalities. Recently, texture analysis of digitized mammograms has shown that high-risk gene carriers and low-risk women have different parenchymal patterns, features that may be useful for identifying women at high risk for breast cancer, thereby intensifying screening protocols and monitoring of treatments for selecting breast cancer patients. As an example, automated mathematical quantitative values can provide information about patterns; high-risk subjects tend to have denser breast tissue as reflected by a negative value in skewness measure, coarser texture of parenchymal patterns as indicated by a higher coarseness measure, a lower fractal dimension measure, a smaller edge gradient measure, and their images tend to be lower in contrast [33].

We found that macular scans of a homogenous group of healthy volunteers of white race without retinal pathology and with similar refractive error demonstrate coherent and consistent quantitative values in all 11 parameters of interest, demonstrating the feasibility of this analysis. This tight clustering is further supported by a consistent correlation of subfoveal choroidal thickness and 3 mm choriocapillaris flow, established with CC imaging parameters, with the textual parameters of interest (Table 1).

Other than computerized texture analysis, other automated and machine-learning analysis of ophthalmic imaging technologies, particularly OCT, have proven promising in terms of identifying pathologies, particularly of the macula [46,47,48]. Thresholding strategies have been the mainstay for quantification of the choriocapillaris. Flow deficit parameters including number, mean size, total area, and density have been used. A systematic review of thresholding studies indicated marked variability of quantification methods [49].

The most used algorithm to analyze CC flow deficits using thresholding is Phansalkar’s method in ImageJ software (National Institutes of Health, Bethesda, MD, USA) [46]. Since its introduction, several modifications have been proposed to adopt this algorithm for analysis of the CC [50,51,52]. Furthermore, binarization of the CC using the Fuzzy C-means method has been used for segmentation and analysis of CC slabs. Chu et al. [53] studied normal adults from all age groups with SS-OCTA using en-face CC images to identify flow deficits with either Standard Deviation from a young normal database as the global thresholding or the Fuzzy C-means method with local thresholding. The authors found a strong correlation and satisfactory agreement between the Standard Deviation and Fuzzy C-means methods. A limitation of the Standard Deviation method is that it requires reference to a normal database for quantification. The advantage of the Fuzzy C-means method is its applicability across different OCTA devices or different scanning protocols; however, the limitations of binarization methods apply to these methods. As new analytic approaches emerge, it has become clear that there is confusion about the appropriate use of thresholding algorithms for quantifying the CC, encouraging new strategies to overcome these deficits and to guide their proper use [52].

Herein, by applying computerized texture analysis to en-face OCTA images of the CC, we propose quantitative standards of CC patterns in healthy eyes and introduce a feasible method for analysis of the CC. Ultimately, we hope the use of this technology will allow for discrimination of flow patterns between healthy eyes versus at risk eyes or eyes with diseased choroids.

To our knowledge, this is the first report of automated complex computerized texture analysis with multiple parameters and algorithms to characterize and automatically quantify texture patterns of the normal CC in OCTA images. Montesano et al. [13] have reported an analysis technique with some similarities to ours. However, their textural analysis focused only on two parameters: an Ising model β and γ, defined as pixel clustering and white pixel density, respectively. Our study differs, in that we provide a greater number of raw parameters for analyzing images of the CC. Additionally, our subjects represent a more homogenous group than those of Montesano et al., whose ages ranged from 16–86, and for whom refractive error was not considered. Possibly due to their heterogeneous study population, one of their two study parameters demonstrated a relatively large standard deviation from the mean value (γ: 0.003 ± 0.012), and thus may prove less meaningful in terms of distinguishing normal from abnormal CC architecture and density.

Use of computerized texture analysis could potentially solve some of the problems encountered by previous researchers attempting to glean meaningful information from OCTA imaging of the CC. For example, work by Wang et al. added to the field by defining normal CC vessel diameter, but their method of analysis requires time and labor-intensive subdivision of CC images into distinct measurement areas, each of which is subsequently analyzed for capillary density (defined as regions that meet a pixel threshold) [12]. Our work offers the advantage of providing normative values regarding CC imaging without depending upon manual extraction of data, while providing textural information beyond the percentage of pixels achieving a certain threshold.

One remarkable finding in our study is that blob size distribution showed a highly concordant homology between all eyes, which may represent a “repeat pattern” of signal density and, thus, perfusion in the superficial CC of healthy young individuals of the same racial background. On this basis, we propose that detecting abnormal patterns of perfusion in this critical retinal layer could improve understanding of chorioretinal disorders and enable detection of early or subtle changes in the CC.

The most significant conclusion from our study stems from the fact that we applied metrics that were all very coherent with consistent quantitative values amongst normal subjects, establishing a “proof of concept” feasibility study of the computerized texture analysis of CC OCTA images. This step is necessary prior to further exploration of clinical applications such as evaluating changes of texture patterns in different clinical settings, investigating its potential value to differentiate texture patterns among pathologies or to monitor progression of texture patterns over time and their clinical correlations. For example, texture analysis may allow us to diagnose a variety of choroidal and retinal diseases before they would otherwise be clinically detectable. Due to the higher-order nature of these parameters that the human eye cannot discern, texture analysis could plausibly lead to earlier diagnosis of ocular disease, thereby improving outcomes. Furthermore, if these textural parameters are established as reliable biomarkers of disease severity, they may be uniquely positioned to monitor disease progression and response to established and emerging therapies.

One limitation of the study is that there are no data on repeatability, reproducibility, and discrepancy, which are important reliability measures [54]; however, our goal in this study was to focus on feasibility rather than reliability. Image averaging techniques are being used more frequently to address discrepancy in particular [55], but despite the substantial advancements in recent years [56], there seems to be a long road ahead to perfection. Another limitation of the study pertains to existing artifacts, which could be confounding factors affecting the analysis. This issue equally limits comparable studies. Due to limitations in depicting vascular flow, deeper layers such as choroidal vessels can be attenuated and have low signal from optical scattering by the RPE. The built-in image processing software in RTVue-XR Avanti system uses thresholding to mask signals it deems to be invalid. In this study, the superficial choriocapillaris (and no deeper vasculature) was analyzed using the automatic CC slab set by the software. Despite all possible limitations including poor penetrance and artifacts, which may affect the sensitivity for signal detection, all subjects were imaged using the same system and analyzed by the same operator with the goal of feasibility testing. Further studies using newer modalities of OCTA with increased penetration and enhanced imaging of deeper structures will lighten this limitation and provide greater insight into the texture analysis of the choriocapillaris and choroidal flow.

Computerized texture analysis of the CC in OCTA images is a promising and novel approach that offers multiple parameters for interpreting differences between images beyond binary analysis of black and white pixels. This method offers a great tool for quantitative analysis that is well-suited for application to larger pools of data and/or numbers of images, as it is less time consuming and more precise than manual extraction of light versus dark binary data. Automated higher-order analysis of blood flow patterns in OCTA images of the CC could also set the groundwork for artificial intelligence technology to be applied in OCTA CC images. Further work is needed to determine how this type of analysis correlates with age, race, and refractive error and, eventually, in pathological states.

In this work, we sought to establish the feasibility of applying advanced mathematical algorithms for texture analysis of CC flow by demonstrating that the various texture parameters yielded consistent values across normal eyes. In future work, we will explore the power of analyzing these texture parameters for differentiating normal and diseased eyes. Thus, we cannot yet speculate which parameter will be most sensitive or important for this task. One great benefit of having multiple measures is that each of these parameters assesses varying textural characteristics of the image, increasing the chances to detect patterns when we apply these measures to diseased eyes. Our primary goal in this phase of analysis was to show similar values for computerized texture parameters within a homogeneous group of normal eyes as the basis for future comparisons between normal and diseased eyes or between different stages of a disease.

As more investigators around the world explore this incredible technology, fascinating applications continue to emerge, helping us to learn about the role of the CC in the pathophysiology, manifestations, and severity of prevalent and debilitating chorioretinal diseases [21,22,23,26,57,58]. Computerized texture analysis of OCTA images of the CC could potentially provide a better understanding of chorioretinal disorders and enable detection of early and subtle changes in the CC relevant to patients’ disease predisposition and progression.

## Figures and Tables

**Figure 1 cells-11-01934-f001:**
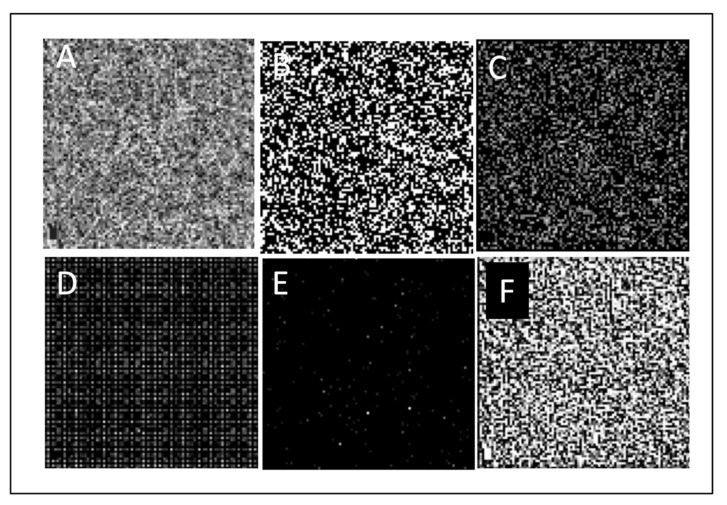
En-face 3 × 3 mm macular OCTA image of choriocapillaris of a young healthy subject (**A**). Texture modification using different filters: Edge filter (**B**), Corner filter (**C**), Histogram of oriented gradients (**D**), Blob localization (**E**), Local binary pattern (**F**).

**Figure 2 cells-11-01934-f002:**
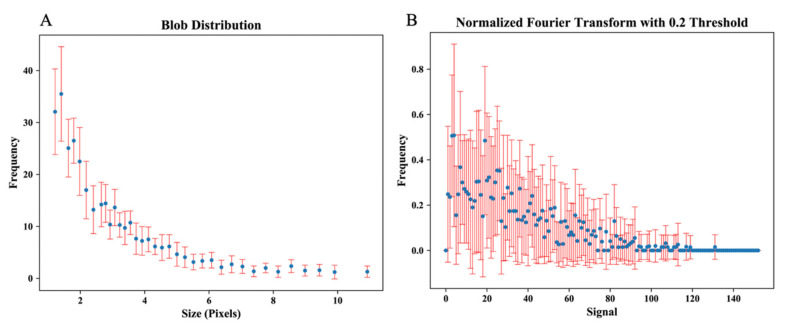
Blob size distribution, each color representing one eye of a subject (**A**). Normalized Fourier transform with 0.2 threshold; each color represents an eye (**B**).

**Figure 3 cells-11-01934-f003:**
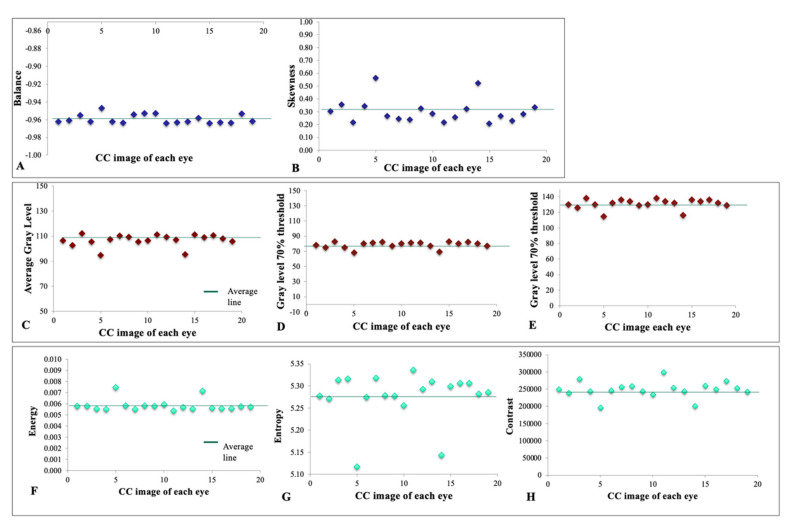
Gray level histogram analysis consisting of measures of Balance (**A**) and Skewness (**B**). Absolute value of gray levels in OCTA images of choriocapillaris in each eye. These set of measures included average gray level (**C**), gray level with 30% (**D**) or 70% threshold (**E**). Spatial relationship among gray levels in OCTA images of choriocapillaris in each eye. Energy (**F**), entropy (**G**), and contrast (**H**) were measured in this category of measures.

**Figure 4 cells-11-01934-f004:**
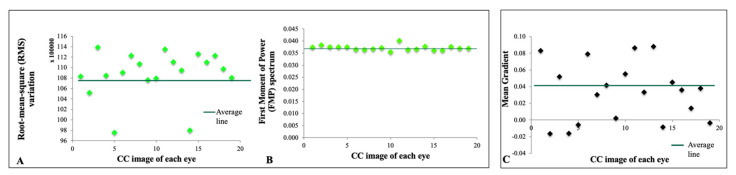
Fourier transform analysis of parameters. Root mean square (RMS) variation (**A**) and First Moment of Power (FMP) spectrum (**B**) is shown. Edge frequency analysis (**C**) is depicted.

**Table 1 cells-11-01934-t001:** Mean and standard deviation from the mean of each texture metric and correlation with established choroidal biomarkers.

Texture Feature	Average Gray Level 30% Threshold	Average Gray Level 70% Threshold	Balance	Skewness	Energy	Entropy	Contrast	Mean Gradient	RMS *	FMP **
**Mean ± SD**	106.80 ± 4.8	130.89 ± 6.3	−0.96 ± 0.005	0.30 ± 0.09	0.0058 ± 0.00053	5.28 ± 0.055	248,156.4 ± 23,327.2	0.033 ± 0.035	16,086,715.5 ± 447,937	0.0371 ± 0.001
**Pearson’s correlation with subfoveal choroidal thickness**	−0.557	−0.625	−0.519	0.010	0.576	0.454	−0.418	−0.402	−0.366	−0.523
**Pearson’s correlation with 3 mm choriocapillaris flow**	0.520	0.556	0.526	−0.442	−0.579	−0.480	0.471	0.599	0.063	0.509

* Root-mean-square variation; ** First moment of power spectrum.

## Data Availability

Data supporting reported results can be found following link: https://docs.google.com/spreadsheets/d/1mT0UIOm7JgIUCyDM9aZnMEvdN5bYc2-0/edit?usp=sharing&ouid=110465867054827951925&rtpof=true&sd=true (accessed on 14 May 2022).

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
