# Peer review of "Computerized Texture Analysis of Optical Coherence Tomography Angiography of Choriocapillaris in Normal Eyes of Young and Healthy Subjects"

_cells, 2022, doi:10.3390/cells11121934_

Round 1

Reviewer 1 Report

In this manuscript, the authors proposed the texture analysis of the CC images taken by optical coherence tomography angiography (OCTA). Even though it might be the first try, unfortunately, in the technical point of view the manuscript is not completed. It just presents the application of the split-spectrum amplitude-decorrelation angiography (SSADA) software to the OCTA images.

  1. Optical coherence tomography (OCT) is an imaging technique based on the optical scattering properties of specimens. To get the specific parameter from the OCT images of different specimens, the gray scale might be adjusted properly. Therefore, it is recommended to add detail explanations about the image processing processes performed to calculate each texture feature.

  1. In section 2, it is recommended to present the en-face OCTA images obtained with the split-spectrum amplitude-decorrelation angiography (SSADA) software. Explanation is necessary on what this software is.

  1. In section 3.1, it would be better to add reference(s) explaining the reason for selecting the parameters. Explanation on the meaning of each texture feature is necessary.

  1. With Figure 3 and 4, the authors conclude that the texture analysis through OCTA gives coherent and consistent results. However, without any control group, it is not easy to get the conclusion. The multiple measurements with a same specimen will give well coherent and consistent results. Similarly, the measurements with a same group will do the same.

Author Response

In this manuscript, the authors proposed the texture analysis of the CC images taken by optical coherence tomography angiography (OCTA). Even though it might be the first try, unfortunately, in the technical point of view the manuscript is not completed. It just presents the application of the split-spectrum amplitude-decorrelation angiography (SSADA) software to the OCTA images.

  • We would like to clarify to the reviewer that SSADA is a vital component of OCTA software, as it decreases the signal to noise ratio of flow detection, enabling visualization of retinal vasculature. Thus, it is an inherent component of OCTA software and not something we have done. Our report is the use of computerized texture analysis on OCTA images of the choriocapillaris.

  1. Optical coherence tomography (OCT) is an imaging technique based on the optical scattering properties of specimens. To get the specific parameter from the OCT images of different specimens, the gray scale might be adjusted properly. Therefore, it is recommended to add detail explanations about the image processing processes performed to calculate each texture feature.

  • In Appendix A, which was included with the original submission, readers are able to see how various texture features were calculated. There is a line referring readers to this supplementary material in lines 104-105.

  1. In section 2, it is recommended to present the en-face OCTA images obtained with the split-spectrum amplitude-decorrelation angiography (SSADA) software. Explanation is necessary on what this software is.
  • We included a figure of the en-face OCTA image at the level of the choriocapillaris in the original submission. It is included in figure 1A. Additionally, we have added a brief explanation of SSADA software in lines 78-79.

  1. In section 3.1, it would be better to add reference(s) explaining the reason for selecting the parameters. Explanation on the meaning of each texture feature is necessary.

  • These are mostly novel parameters that our group is reporting as it pertains to textural analysis of the choriocapillaris in particular. We acknowledge in lines 229-231 that this is the first report of computerized texture analysis used to quantify patterns of the normal choriocapillaris in OCTA images. Therefore, they will be not be extensively reported in similar studies of the choriocapillaris in the literature. However, in lines 180-186 of the original work, we discuss prior instances where entropy, energy, and contrast parameters have been reported. We further discuss a similar previous study by Montesano et al. to provide context regarding the parameters we have chosen and their implications for texture analysis of the choriocapillaris. Furthermore, we would direct the reviewer to descriptions of these texture parameters and how they were calculated that were included in the original submission on lines 87-94, as well as on the supplementary file included.

  1. With Figure 3 and 4, the authors conclude that the texture analysis through OCTA gives coherent and consistent results. However, without any control group, it is not easy to get the conclusion. The multiple measurements with a same specimen will give well coherent and consistent results. Similarly, the measurements with a same group will do the same.

  • We must emphasize that this study is a proof-of-concept demonstrating a novel application of textural analysis of the choriocapillaris layer. We do not intend for there to be a control group due to this point nor should one be expected in a proof-of-concept study design. The conclusion that we have derived coherent and consistent results in our initial patient population is an important one in this proof-of-concept image analysis study. we respectfully disagree with the notion that measurements from the same group will always be coherent and consistent, since this should not be expected to be the case. We have observed tight clustering of textural parameters, supporting the conclusion that texture analysis of the choriocapillaris is feasible and can be extended into different patient populations as next steps for application. If we observed, instead, that there was a great deal of variation in healthy eyes, it would preclude us from extending our work to study eyes with disordered choroids/choriocapillaris. Our results and conclusion are important in light of this.

Reviewer 2 Report

The Comments were addressed sufficiently. However, it is worthy to cite some of the papers, which address about the feature characterization using OCT.

Ex: Biophotonic approach for the characterization of initial bitter-rot progression on apple specimens using optical coherence tomography assessments....

Author Response

The Comments were addressed sufficiently. However, it is worthy to cite some of the papers, which address about the feature characterization using OCT.

Ex: Biophotonic approach for the characterization of initial bitter-rot progression on apple specimens using optical coherence tomography assessments.

Thank you for your suggestion. We have added four papers, including the one suggested, displaying the use of OCT and OCTA for feature characterization in diverse fields in both biology and medicine. We hope that these will further anchor our manuscript in the current literature and further demonstrate the feasibility of our work.

Reviewer 3 Report

Dear Author(s),

After the due revision, I would now recommend this paper for publication. Congratulations!

Author Response

Thank you for your comments and suggestions throughout the review process!

This manuscript is a resubmission of an earlier submission. The following is a list of the peer review reports and author responses from that submission.

Round 1

Reviewer 1 Report

Dear Author(s),

After the due revision, I would now recommend this paper for publication. Congratulations!

Reviewer 2 Report

In this manuscript, the authors proposed the texture analysis of the CC images taken by optical coherence tomography angiography (OCTA). Even though it might be the first try, unfortunately, in the technical point of view the manuscript is not completed. It just presents the application of the split-spectrum amplitude-decorrelation angiography (SSADA) software to the OCTA images.

  1. Optical coherence tomography (OCT) is an imaging technique based on the optical scattering properties of specimens. To get the specific parameter from the OCT images of different specimens, the gray scale might be adjusted properly. Therefore, it is recommended to add detail explanations about the image processing processes performed to calculate each texture feature.

  1. In section 2, it is recommended to present the en-face OCTA images obtained with the split-spectrum amplitude-decorrelation angiography (SSADA) software. Explanation is necessary on what this software is.

  1. In section 3.1, it would be better to add reference(s) explaining the reason for selecting the parameters. Explanation on the meaning of each texture feature is necessary.

  1. With Figure 3 and 4, the authors conclude that the texture analysis through OCTA gives coherent and consistent results. However, without any control group, it is not easy to get the conclusion. The multiple measurements with a same specimen will give well coherent and consistent results. Similarly, the measurements with a same group will do the same.

Reviewer 3 Report

The given study was carried out in order to derive results for the performance of Computerized Texture Analyzing (CTA) technology in derivation of Optical Coherence Tomography Angiography (OCTA) in the field of biomedical imaging as opposed to angiography technologies utilizing dyes such as Fluorescein and Indocyanine, particularly in the context of exploring the degeneration of the Choriocapillaris CC, in different patients. This was done in order to refine the grade of the images obtained to further expand the scope of medical imaging in order to make convenient for analysts to study the (CC) and any trends shown in the different blood flow characteristics that occur within the eye of a healthy human so that sight related conditions and their effects can be understood more coherently.

The paper needs to be properly enhanced with the given comments.

Comment1: Despite the advancements in the technologies used in the field of biomedical imaging, especially in the context of vision, commonly utilized technologies such as and In-vivo technologies such as angiography and dye-less, in-vivo, non-invasive methods like OCTA, are not without their shortcomings. Typical angiography technologies, have been limited in their reach due to the fenestration of the CC vasculature, that is, in this context the arrangement and openings of the membrane which causes leakage of dye. In the case of the more advanced OCTA, the main drawback lies in the images obtained themselves, with there being little discrimination of dark and light pixels which in essence being a binary differentiation of values, reduces the clarity of the processed images.

However, even when CTA is applied to the OCTA process to En-face the imaging, via complex mathematical manipulations and sophisticated algorithms for monitoring various metrics of the image, the obtained, En-faced images which contain a large amount of data supplied by complex image processing of the CTA, are not fully interpretable by the current imaging standards which are limited to the black and white pixel density, which leaves a large portion of the data unanalyzed or even simply cut-off at the final renderings.

 Given the analyzing capabilities of CTA, which produces a vast scale of information that can be processed in the En-faced version of the OCTA image, that in theory produces a more clearer image due to the greater discrimination between light and dark pixels, it can be said that the underlying deficit falls on the abilities of the current ‘imaging’ systems due to their rather limitative pixel density which hinders details being produced on final images as the flow voids that denote microscopic structural changes and data deficits are interpreted and quantified in either black or white.

Can authors comment on this point?

Comment 2: As it can be seen, the biggest hindrance to the analysis of minute details is the imaging system itself, as such further improvements can be made to the imaging systems so that they have a higher discrimination between light and dark pixels as the higher variation in light levels of the pixels would allow interpretation of flow voids and data deficits based on the intensity or reflection energy of the signal which varies with different microscopical structures, this would produce an En-faced image that would be more detailed containing visual information of different surfaces and structures that were encountered by the sensor which would allow analysts to more efficiently analyze the biological structure.

Comment 3: Texture analysis on digitized mammograms has revealed that high-risk gene carriers and low-risk women have different parenchyma patterns, recognizing
features useful for identifying women at high risk for breast cancer, intensifying
screening protocols, and monitoring breast cancer patients' treatment.

Comment 4: OCTA uses light waves rather than sound waves, medium opacity can interfere with optimum imaging. As a result, the OCTA will be restricted in cases of
vitreous hemorrhage, thick cataract, or corneal opacities.

Comment 5: OCTA measures retinal thickness by using light beams. This test uses no radiation or X-rays, and an OCTA scan is not painful or unpleasant. But if the
patient faced this test regularly, it may damage to the cells in the eye.

Comment 6: It’s better to understand if there’s the graphical structure of the experiment.

Comment 7: There’s several grammatical mistakes but that mistakes are not that barriers to refer the article. But it is strongly recomended to Proof read the paper.

Comment 8:   Line 183,  Aside should be apart

Comment 9: Line 184, analyses should be analysis